# Analysis of Influencing Factors of Apathy in Patients with Parkinson’s Disease

**DOI:** 10.3390/brainsci12101343

**Published:** 2022-10-04

**Authors:** Ruirui Luo, Yumeng Qi, Jiuqin He, Xiaoqi Zheng, Wenhua Ren, Ying Chang

**Affiliations:** 1Department of Neurology, China-Japan Union Hospital of Jilin University, Changchun 130033, China; 2School of Statistics, East China Normal University, Shanghai 200050, China

**Keywords:** apathy, impulse control disorder, Parkinson’s disease, REM sleep behavior disorder (RBD)

## Abstract

Background: Apathy is a common non-motor symptom of Parkinson’s disease (PD). The influencing factors of apathy are currently controversial. This study aimed to describe the clinical characteristics of PD-associated apathy and to analyze the associated risk factors. Methods: Two hundred patients diagnosed with PD were selected. Included patients were divided into an apathetic group and a non-apathetic group. Demographic and clinical data, motor symptoms, non-motor symptoms and medication use of the two groups were assessed. Results: The incidence of apathy was 69%. Demographic and clinical data, motor symptoms, non-motor symptoms and medications use were statistically significant. Conclusions: PD patients with more severe motor symptoms, cognitive impairment, depression, anxiety, RBD, excessive daytime sleep, fatigue, low education level, long disease course, poor quality of life and lower DA dosage are more prone to apathy. Cognitive function, quality of life, educational level, DA and LEDD are independent risk factors for apathy.

## 1. Introduction

Parkinson’s disease (PD) is the second most common neurodegenerative disease, with typical motor symptoms such as bradykinesia, resting tremor, muscle rigidity, and postural disturbances [1,2]. However, with multiple neurotransmitter systems and neural circuits affected, PD patients also have a wide range of non-motor symptoms [3]. Apathy is a common non-motor symptom and is also the most common neuropsychiatric symptom in patients with PD [4]. Apathy is often confused with depression due to its non-specific clinical manifestations, which leads to an insufficient understanding of apathy and makes early identification difficult. However, apathy also has a serious impact on the quality of life of patients and caregivers [5], and even has a high disability rate. Studies have shown that PD patients with apathy had more severe motor and executive dysfunction, and an even a higher risk of dementia than patients without apathy [6], suggesting that apathy is a crucial factor that may accelerate the progress and deterioration rate of PD patients. Previous studies have shown that the occurrence of apathy in PD patients was influenced by a variety of factors. In terms of demographics, studies have shown that men, older adult patients, and patients with low levels of education are peculiarly prone to apathy [7]. However, other authors have concluded that apathy is not associated with either gender or age [8]. Motor symptoms have also been associated with PD apathy, which tends to be associated with a higher Uniform Parkinson Disease Rating Scale (UPDRSIII) score [9]. However, studies have also shown that movement disorders cannot predict the development of apathy [10]. Prior studies have shown that low doses of levodopa increase the risk of apathy in patients with PD [11], while Brown found no relationship between levodopa doses and apathy [9]. Regarding non-motor symptoms, the relationship between apathy and impulse control disorders (ICDs) is still unclear. In addition, a few studies have investigated the association of apathy and REM sleep behavior disorder (RBD), and there is a lack of research on excessive daytime sleepiness (EDS) and apathy. Current domestic and international research results about risk factors for apathy in PD are inconsistent, and further research is still needed to understand the risk factors associated with apathy and possible relationships between factors. Therefore, this study aimed to evaluate the demographic and clinical characteristics of apathy in PD patients and to explore related risk factors. Therefore, this study aimed to evaluate the demographic and clinical characteristics of apathy in PD patients and to explore related risk factors.

## 2. Materials and Methods

### 2.1. Study Design and Sample

In this cross-sectional, retrospective study, a total of 200 patients with PD who were treated at the China-Japan Union Hospital of Jilin University from September 2019 to December 2021 were enrolled. Inclusion criteria were: (1) All patients must meet the PD diagnostic criteria established by the Movement Disorder Society (MDS), as previously described [12]; (2) Patients were permitted to cooperate with research assistants to complete the scale assessment. Exclusion criteria were: (1) Parkinson’s syndrome caused by intracranial infection, cerebrovascular disease, poisoning, trauma, drugs, etc.; (2) Patients with Parkinson plus syndrome such as progressive supranuclear paralysis, multiple system atrophy, cortical basal ganglia degeneration, Lewy body dementia, etc.; (3) Patients with other neurological disorders, psychiatric disorders, etc.; (4) Patients who were not able to complete scale assessments for various reasons.

The apathy evaluation scale (AS) was used to group the enrolled patients. AS is recommended by the International Association of Movement Disorders or Movement Disorder Society (MDS) for the assessment of apathy in patients with PD [13]. The scale consists of 14 questions with a total score of 42 points, and a total score of greater than or equal to 14 points to diagnose apathy, with higher scores indicating a higher, more serious degree of apathy. Based on the scores, the patients were divided into the apathetic and non-apathetic groups. The assessments were made under the off-period state, then clinical data were collected.

### 2.2. Data Collection and Assessment

Demographic data collected included sex, age, age of onset, education level, and disease course. Clinical data collection: Motor symptoms assessment included: the third part of the Unified Parkinson’s Disease Rating Scale of the Movement Disorder Society (MDS-UPDRS-III) and the Hoehn and Yahr grading of stages (H-Y grading), and freezing of gait (FOG) questionnaire. General global cognition assessment used the Montreal Cognitive Assessment (MoCA) scale, in which trail making test, cube copy, clock drawing test, language fluency, and abstraction were used for executive function assessment. Other non-motor symptom assessment included: Hamilton Depression Scale (HAMD), Hamilton Anxiety Scale (HAMA), the Epworth sleepiness scale (ESS), REM sleep behavior disorder (RBD) survey questionnaire, fatigue severity scale (FSS), Impulsive and Obsessive-Compulsive Control Disorder Questionnaire (QUIP Questionnaire), and Parkinson’s Disease Quality of Life Questionnaire (PDQ-39).

### 2.3. Statistical Analysis

All statistical analyses were performed using SPSS 22.0 (IBM SPSS, Chicago, IL, USA). Two major tests involved in the study were univariate analysis and multivariate analysis. The first test served to exam if one variable had a different distribution between the apathy and non-apathy group. Pearson’s chi-squared test was applied for categorical data, and Mann–Whitney U Test, a preferred nonparametric test for non-normal distribution data, was performed for the numerical data. The level of statistically significant difference was established as *p* < 0.05.

Based on the results of univariate analysis, the study moved to multivariate analysis, that is, stepwise regression analysis. It aimed to quantify the relationships between important variables and apathetic status, and also to find risk factors. To avoid multicollinearity, a problem might occur when two independent variables were highly correlated and would distort the interpretation of the relationships among variables, Spearman’s correlation coefficients were calculated pairwise. If any pair had a coefficient larger than 0.8, then one of the variables would be dropped base on IV (information value), a technique helped determine the predictive power of one variable on the dependent variable, that was apathetic status in this study. After that, a stepwise regression model, an efficient way to find out the best combination of variables, was constructed. The criteria for model selection was the Akaike information criterion (AIC), an estimator of prediction error and thereby relative quality of statistical models for a given set of data [14].

Traditionally, the model could only include statistically significant variables. However, it might miss essential features, and that was underfitting. AIC dealt with both the risk of underfitting and overfitting.

AIC: origin from information theory, which estimates the relative amount of information lost by a given model; a better model should lose less information

## 3. Results

### 3.1. Univariate Analysis of Factors Associated with Apathy in PD

A total of 200 patients were enrolled in the study, with 138 apathetic PD patients. Incidence of apathy was 69%. Figure 1 presents the relative values of mean and median for all demographic, clinical and motor, and non-motor factors, giving a general picture of values of interest in apathetic and non-apathetic groups. Table 1 illustrates the results of univariate analysis of demographic data, motor, non-motor, and medication factors associated with PD apathy.

Significant differences were found between the apathetic and non-apathetic groups in the distribution of education level and disease duration (*p* < 0.001, *p* = 0.0143).

Among all of the motor symptoms listed, UPDRS III, H-Y grading, and FOGQ showed statistical differences between the two groups (*p* < 0.001, *p* = 0.0038, *p* = 0.0063). For non-motor symptoms, only ICD failed to reject the null hypothesis under the significance level of 0.05 (*p* = 0.5957, *p* = 0.0638), indicating no statistical differences were found in the distribution of these factors between the two groups.

For medication use, the distribution of dopamine receptor agonists (DA) was significantly different between the apathetic and non-apathetic groups (*p* = 0.0098).

### 3.2. Multifactorial Analysis of Parkinson’s Disease Apathy

The study applied Spearman’s rank correlation coefficient and any pair of variables had coefficient over 0.8 was considered highly correlated. The figure for age and age of onset was 0.89, and for HAMA and HAMD was 0.84 (Figure 2). All other variables had coefficients lower than 0.8. Information value (IV), a widely applied index for evaluating factor importance, was calculated (Table 2) to select one variable out of two. The higher the IV value, the more important the variable is. Age and HAMD had larger values to be included in modeling, while age of onset and HAMA were excluded.

Table 3 demonstrates the results of multivariate stepwise logistic regression analysis. This model was realized by adding or dropping variables of interest bidirectionally based on the criterion to minimize information loss. Five factors—education level, MoCA, PDQ-39, DA and LEDD—were selected, with the minimum AIC being 219.2.

The sign and scale of coefficients accounted for the relationships between predictive variables and dependent variables. The signs of PDQ-39 and LEDD were positive, suggesting that higher values of these factors increased the risk of PD apathy.

Consequently, negative signs included education level, MoCA, dopamine receptor agonists, and which had a negative influence on the risk of PD apathy. The coefficient scales of medications factors are less than other variables, indicating a smaller influence on PD apathy.

## 4. Discussions

In the present study, the incidence of apathy was 69%. Univariate analysis of multiple factors related to PD revealed that education level, disease duration, motor symptoms, and most non-motor symptoms were distributed differently between the apathy and non-apathy groups. After adjusting for covariates, multivariate stepwise logistic regression revealed that the PD patients with apathy had lower educational levels, higher PDQ-39 scores, higher levodopa equivalent daily dosage, lower MoCA scores, and lower DA dosage.

Studies in Taiwan, Europe, and the United States have reported that the incidence of apathy in PD is widely distributed from 17–70% [15,16]. The incidence of apathy in this study is 69%, which is a high incidence. It may be that patients in this area are constrained by economic and cultural levels. Most patients are not in the early stage of the disease when they receive treatment, and patients with a longer course of the disease, such as in the advanced stages, are more likely to be accompanied by apathy, which leads to a high incidence.

In terms of demographic data, previous studies have suggested that apathy in patients with PD is associated with gender, age, and educational level, and is most likely to occur in males, those of an advanced age, and those with a lower level of education [17]. The results of the present study are similar to previous studies reported by Cubo et al. [7] and Gorzkowska et al. [8]—showing that apathy is associated with a low educational level. Cognitive function is more likely to be affected in patients with low education levels, suggesting that more learning may help to improve PD apathy. Moreover, Gorzkowska et al. [8] offered a different view—lower levels of education may be the result of apathy rather than the cause. However, regression analysis in a cross-sectional study cannot infer causality. Further study is needed.

The present study also indicated that apathy may be associated with disease duration. Univariate analysis revealed that the range of disease duration in the non-apathy group was wider and the median was smaller. Patients with a longer disease course may have a greater chance of developing apathy. In fact, a long course of the disease is more often accompanied by more severe motor symptoms and non-motor symptoms, as reported in domestic and foreign studies as being closely related to the occurrence of apathy. Therefore, patients in the advanced stages are susceptible to apathy. Gender, age, and age of onset seemed to have little connection with apathy in the present study. More studies with larger samples are needed to further clarify the impact of demographic data on the apathy of PD patients.

Motor symptoms are the most intuitive clinical manifestation in patients with PD. Previous studies have reported that apathy was associated with the severity of movement disorders [18,19,20], including patients with higher UPDRSIII scores, who tended to have higher AS scores. The present study also showed a statistically significant difference in UPDRSIII scores, H-Y grading, and FOGQ score between the two groups. This may possibly be explained by the fact that, on the one hand, motor symptoms such as bradykinesia, resting tremor, and muscle rigidity cause a state of indifference to external affairs associated with being disabled in action; on the other hand, the lack of dopamine neurotransmitters may account for it. It is recognized that apathy is related to dopaminergic impairment affecting motivational behavior, and other studies have found that motor symptoms were associated with apoptosis of dopaminergic neurons in substantia nigra pars compacta, suggesting that the two may overlap in pathogenesis.

The results of the present study have shown that the difference in MoCA scores between the two groups was statistically significant. The collective evidence previously suggested that apathy was closely associated with cognitive decline [21,22]. A meta-analysis by D’Iorio et al. [23] found that apathy in patients with PD was associated with severe cognitive impairment, particularly with changes in the prefrontal subcortex leading to executive dysfunction. Multiple studies have confirmed that apathy is associated with executive dysfunction [24,25], which may be owed to both having the same neural circuit damage and transmitter abnormalities. Neuroanatomy suggests that executive dysfunction in patients with PD is primarily associated with abnormalities in the frontal-striatal pathway, while decreased midbrain dopaminergic levels and abnormalities in norepinephrinergic and cholinergic in terms of neurotransmitters are also thought to be associated with executive dysfunction [26]; additionally, previous studies found that the pathogenesis of apathy was associated with the damage of the prefrontal-basal ganglia loop that mediates the motivational loop [27], and related imaging studies have further confirmed that apathy was associated with the destruction of specific medial frontal cortex and subcortical structures, including the anterior cingulate cortex (ACC), medial orbitofrontal cortex (OFC), and ventral striatum (VS) [28,29]. The changes of neurotransmitters also play an important role in the pathophysiology of apathy. In addition to the recognized dopaminergic impairment affecting motivational behavior, the impairment of non-dopaminergic pathways, such as serotonin, norepinephrine, acetylcholine, and adenosine were also associated with apathy [30]. Therefore, there is an overlap in the pathogenesis of apathy and executive dysfunction. The results of the present study have indicated that apathy is related to cognitive function, and correlation was also found between apathy and executive dysfunction, which is consistent with previous studies. Based on these results, we speculate that early cognitive function training or medication intervention may improve cognitive function, which may be helpful for relieving apathy.

Depression is a common non-motor symptom similar to apathy, and previous opinions widely considered that apathy is a companion symptom of depression. However, as further studies about apathy have been published, it has gradually become clear that apathy is a separate non-motor symptom that can be distinguished from depression [31,32,33]. In addition to the independent relationship between depression and apathy, studies have also shown a link between them. The difference in HAMD scores between the two groups was also statistically significant in the present study. HAMD may be involved with the orbitofrontal lobe and anterior cingulate cortex, amygdala, thalamus, and ventral striatum, which are also associated with the pathophysiology of both depression and apathy [29]. Therefore, it is necessary not only to distinguish and identify potentially apathetic patients clinically as soon as possible during the early stage, but also to prevent diagnostic omissions. The present study found that the difference in HAMA scores between the two groups was also statistically significant, which may be related to the overlap of pathogenesis between the two. It has been reported that the pathogenesis of anxiety and apathy in patients with PD is associated with abnormal pathways in the striatum nigra and nigra limbic system [34], while neurotransmitter disruption of the dopaminergic and serotonergic systems is common in the pathogenesis of anxiety and apathy [35]. Thus, the overlap in the pathogenesis of anxiety and apathy needs further exploration, which may be an ideal way to target therapy.

A recent study has found that patients with idiopathic RBD had local cerebral blood-flow abnormalities, and these abnormal areas are mainly functional brain regions in the marginal structure that are associated with motivation and mood [36]. Therefore, some researchers have speculated about whether the occurrence of apathy is associated with RBD. Studies by Bargiotas et al. [37] found that patients with PD who have RBD had a higher probability of developing apathy and higher AS scores, thus indicating that RBD may be a potential risk factor for apathy. This is consistent with the conclusion of Mahmood et al. [38]. Because it is relatively easy to identify RBD in the early stages of the disease or even in the prodromal stage, its presence may facilitate early clinical identification of apathetic patients. Therefore, it is beneficial for patients with RBD to be monitored and screened. The results of the present study showed that differences in RBD scores between the two groups were statistically significant, and that the RBD score of the apathetic group was significantly higher than that of the nonapathetic group, which is consistent with the results of the above previous studies.

EDS is also a common non-motor symptom in patients with PD, marked by a state of indifference to external things. The collective evidence indicates that dopamine plays an important role in maintaining wakefulness [39], while PD apathy has been thought to be associated with a dopaminergic neuronal defect, which results in a hypothesis that apathy is associated with EDS to some extent. At present, studies are lacking on the correlation between EDS and apathy. The present study showed that that the differences in EDS were statistically significant; however, further research is needed to verify the relationship between the two. Consistent with the conclusion of a previous study [27], the present study found that patients with a poor quality of life were more likely to develop apathy, and further analysis indicated that it was an independent risk factor for apathy, suggesting that the two affect each other and may have a vicious circle trend. Therefore, interventions for apathy have a significant effect on improving the patients’ quality of life.

Fatigue is also a common non-motor symptom in patients with PD, and many patients consider it the most disturbing symptom. Extensive studies have shown that PD patients with fatigue are more likely to develop apathy, which may be related to the dopaminergic neuron defects in the pathogenesis of both fatigue and apathy [40]. Siciliano et al. [41] found that apathy was the main baseline variable that predicted fatigue severity in PD patients. Serotonin and norepinephrine are involved in fatigue and apathy, and, anatomically, both are thought to be associated with prefrontal-basal ganglia circuit damage [42]. The results of the present study also support the conclusions of previous studies that the difference in FSS scores between the apathetic and non-apathetic PD patients is statistically significant, and the FSS scores in the apathetic group are higher than those in the nonapathetic group, indicating that patients with severe fatigue are more likely to develop apathy.

Being similar to apathy, ICD is also associated with motivation and reward systems mediated by the dopaminergic system; apathy is a symptom of low dopamine while ICD is high [43]. However, Scott et al. [44] showed that apathy and ICD also coexisted in more than one-third of patients with PD. In addition, numerous studies have found that patients with ICD have higher AS scores [45]. New-onset ICD may occur after DBS [46]. ICD may occur in patients not taking dopamine agonists or taking high-dose dopamine agonists. The mechanisms by which ICD and apathy coexist in patients with PD are unclear and cannot be explained only by the dopaminergic system. Similar neuromodulation network disruptions may exist between apathy and ICD, or both may be involved with other neurotransmitters [47]. In the present study, no correlation was found between ICD and apathy. They are both related to motivational disorders in terms of symptoms and pathogenesis; thus, it may be of great significance for the prevention and treatment of apathy and ICD if the relationship between the two could be clarified.

Levodopa and dopamine receptor agonists are the basic medications for patients with PD. However, in the present study, univariate analysis indicated only DA had effect on apathy, LEDD showed little difference between the two groups. Starkstein et al. [48] showed that apathy was associated with dopaminergic drugs such as DA, and administration of these drugs can improve apathy in patients with PD. As D3 receptors are expressed primarily in the marginal region and have high levels of expression in the ventral striatum (including the nucleus accumbens), the occurrence of apathy is associated with damage to motivational loops mediated by the ventral striatum, and D3 receptor agonist drugs can be used to improve PD apathy [49]. Nevertheless, Rosqvist et al. [50] found that dopaminergic drugs aggravated symptoms of apathy in PD, including in patients who responded poorly to dopaminergic drugs. Chung et al. indicated that no correlation was found between apathy and dopaminergic drugs, speculating that apathy may be associated with striated extracorporeal lesion rather than striatum dopaminergic defects [51].

Multivariate stepwise logistic regression analysis indicated that cognitive function, quality of life, educational level, DA, and LEDD are independent risk factors for apathy. Clinically, the improvement of educational level, cognitive function, quality of life, and the rational use of drugs may relieve the symptom of apathy. Specifically, among the five factors, the coefficient of education level is maximal, which indicated that educational level had a significant impact on apathy; therefore, we should pay attention to it.

## 5. Limitations

The present study also has certain limitations, including that it is a single-center retrospective study with a relatively small sample, which limits the generalizability of the results to other populations or locations and limits the extent of follow-up because of using secondary data within a given time period. The small sample also limited the data relating to surgical solutions, so we did not include STN DBS surgery as a variable, which is included in many previous studies.

## 6. Conclusions

With many influencing factors and complex relationships between them, PD patients with low educational attainment, severe motor symptoms, cognitive impairment, executive function disorder, depression, anxiety, RBD, fatigue, and lower DA are more likely to develop apathy. Independent risk factors for apathy include cognitive disorder, quality of life, educational level, DA, and LEDD. Clarifying the relationship between these factors and apathy may enable early screening and intervention in PD patients with high risk of apathy, delaying or even preventing the occurrence of apathy.

## Figures and Tables

**Figure 1 brainsci-12-01343-f001:**
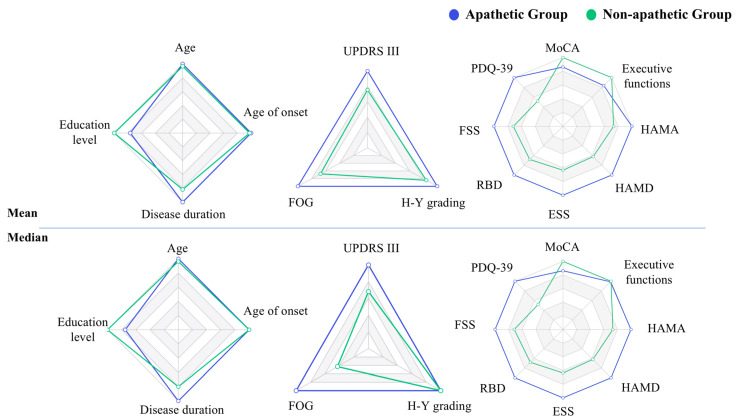
Radar maps of mean and median.

**Figure 2 brainsci-12-01343-f002:**
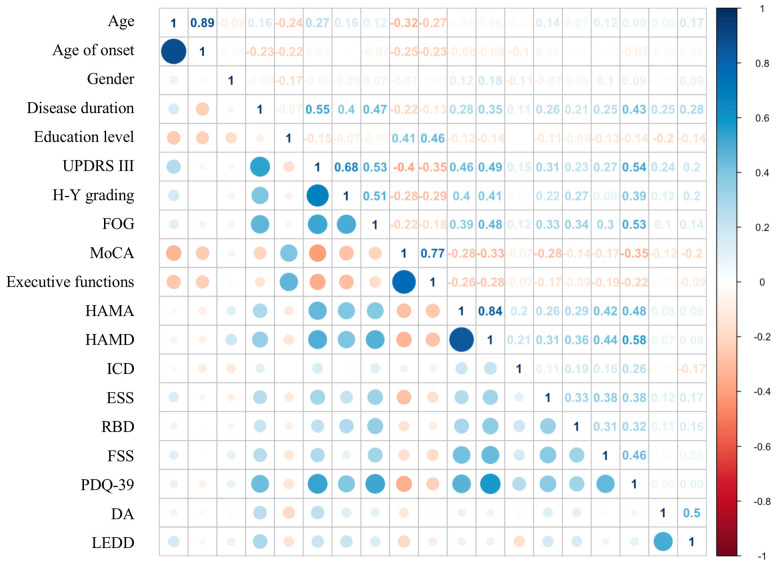
Correlation plot.

**Table 1 brainsci-12-01343-t001:** Univariate Analysis: Comparison of apathetic and nonapathetic groups.

	Apathetic Group (*n* = 138)	Nonapathetic Group (*n* = 62)	p−Value
Demographics Characteristics			
Gender			0.6770
Male	63 (45.65%)	31 (50%)	
Female	75 (54.35%)	31 (50%)	
Age (years)	65.00 (58.00, 69.00)	62.50 (53.50, 69.75)	0.0857
Age of onset (years)	58.00 (53.00, 65.00)	58.00 (49.25, 65.75)	0.2586
Education level (years)	9.00 (7.25, 12.00)	12.00 (9.00, 15.00)	<0.001
Disease duration (years)	5.00 (3.00, 7.00)	4.00 (2.00, 7.00)	0.0143
Motor Symptoms			
UPDRS III	47.00 (33.25, 64.00)	32.00 (22.25, 54.00)	<0.001
H-Y grading	2.00 (1.50, 3.00)	2.00 (1.00, 2.00)	0.0038
Freezing of gait	7.00 (2.00, 13.00)	3.00 (1.00, 8.50)	0.0063
Non-motor Symptoms			
MoCA	22.00 (17.25, 24.00)	24.00 (22.00, 26.00)	<0.001
HAMD	15.00 (9.00, 23.00)	9.00 (5.00, 14.25)	<0.001
HAMA	11.00 (6.00, 17.00)	8.00 (4.75, 12.25)	0.0037
ESS	5.50 (1.00, 9.00)	2.00 (1.00, 7.00)	0.0042
RBD	3.00 (1.00, 7.00)	1.00 (0.00, 4.00)	0.0114
FSS	40.00 (20.50, 54.00)	22.00 (11.25, 40.75)	< 0.001
Executive functions	5.00 (3.25, 6.00)	6.00 (4.25, 7.00)	0.0007
ICD			0.0638
Yes	20 (14.49%)	4 (6.45%)	
No	118 (85.51%)	58 (93.55%)	
PDQ-39	44.50 (26.25, 61.00)	20.00 (8.00, 33.75)	<0.001
Medication Taking			
LEDD	400.00 (281.20, 593.80)	300.00 (0.00, 534.4)	0.3420
Dopamine receptor agonists	0.00 (0.00, 75.00)	0.00 (0.00, 75.00)	0.0098

Notes: Probability presented for qualitative variables. Median and interquartile range for quantitative variables. SD is shorted for standard deviation.

**Table 2 brainsci-12-01343-t002:** Information value.

Variables	IV
Disease duration	0.7475
MoCA	0.7375
Age	0.6570
HAMD	0.6068
Age of onset	0.5604
Freezing of gait	0.4932
PDQ-39	0.4926
UPDRSIII	0.4116
FSS	0.4049
HAMA	0.3611
Education level	0.3501
LEDD	0.3443
ESS	0.2844
RBD	0.2542
H-Y grading	0.2001
ICD	0.1309
Dopamine receptor agonists	0.1016
Gender	0.0076
Executive functions	0.0056

Notes: Variables are arranged in descending order of IV (information value), a numerical value that quantifies the predictive power of independent variables in capturing the binary dependent variable.

**Table 3 brainsci-12-01343-t003:** Multivariate analysis: Logistic regression of influencing factors of apathy with PD.

	Coefficients	SD	p−Value	Odds Ratio	95% Confidence Interval
(Intercept)	0.7475	0.1631	<0.001	2.1118	(1.5341, 2.9069)
Education level	−0.0153	0.0056	0.0065	0.9848	(0.9741, 0.9956)
MoCA	−0.0091	0.0061	0.1397	0.9909	(0.9791, 1.0029)
PDQ-39	0.0057	0.0012	<0.0001	1.0057	(1.0033, 1.0080)
Dopamine receptor agonists	−0.0012	0.0006	0.0276	0.9987	(0.9976, 0.9999)
LEDD	0.0003	0.0001	0.0058	1.0003	(1.0001, 1.0005)

Notes: Variables are selected by stepwise logistic regression which minimized Akaike information criterion (AIC). SD is shorted for standard deviation. Confidence interval is calculated for odds ratio.

## Data Availability

The data was not publicly available due to university policies.

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
