# Peer review of "Analysis of Influencing Factors of Apathy in Patients with Parkinson’s Disease"

_brainsci, 2022, doi:10.3390/brainsci12101343_

Round 1
Reviewer 1 Report
This paper aims to ”describe the clinical characteristics of PD-associated apathy and to analyze the associated risk factors". My concern is that these aims are ill defined and the method chosen is not appropriate. The authors used data from 200 PD patients, each with multiple parameters, separated the data to PD with and without apathy, and compared between the groups using univariate and correlation analyses. Due to the expected coupling between the multiple parameters, I doubt what the meanings of the results are. As clearly stated by the authors, their analyses cannot predict causality. However, without causality it is difficult to understand what the results could mean. Is longitudinal data of these patients is accessible? This would help to predict causality and interactions between parameters. Besides that, to convince us with the correctness of the results, I suggest to repeat all analyses but this time to separate the data into two groups by a different parameter for example depression. If the results will be significantly different, it will suggest that the linear analysis used has meaning. To study apathy I suggest first to consider apathy patients without PD. Comparison of such patients with patients having PD with apathy will enable to characterize the uniqueness of PD-apathy patients. At a second step, comparison with PD without apathy will help to define apathy in PD.
Specific comments:
· There is no hypothesis – for a pure exploratory study, the sample is too small.
· There are no references for the tests used to obtain patent's parameters.
· Measurements were performed at the OFF state. How this state was achieved?
· Method session in too short. It is written for statisticians not for conventional readers. Please explain and give references to all used analyses. For example, why these statistical tests were chosen, how you defined cutoff.
· Please explain the logic and give details for the stepwise regression. Why 5 factors were used and not less or more?
· Did you test dependency between parameters?
· Figure captions are missing for all figures.
· Discussion does not add much for understanding. This is also because there is no hypothesis. I expect more direct reference of the findings to previous results that would lead to some understanding.
Author Response
Point 1: There is no hypothesis – for a pure exploratory study, the sample is too small.
Response1: This study was a retrospective case-control study, the numbers of cases was almost all of the patients admitted to our hospital in the last two years.
Point 2:There are no references for the tests used to obtain patent's parameters
Response2: The tests and the scales applied on the study are consistent with the criteria of the international movement disorder society(MDS).
Point3: Measurements were performed at the OFF state. How this state was achieved?
Response 3: Thanks for your reminding, we ask for the patients to diacontinue the medication for more than 4 hours and assess the motor scores of UPDRS-III to ensure the drug-off state, and luckly, most of the patients could estimate accurately when iwas his/her off state.
Point 4:Method session in too short. It is written for statisticians not for conventional readers. Please explain and give references to all used analyses. For example, why these statistical tests were chosen, how you defined cutoff.
Response 4: Thanks for your reminding ,we have reveised the section of methods.
Point 5:Please explain the logic and give details for the stepwise regression. Why 5 factors were used and not less or more?
Response 5: We took all the factors to the regression and finally retained five factors after removing the interaction of each other.
Point 6: Did you test dependency between parameters?
Response 6: We applied the stepwise statistic methods to remove the dependency between parameters.
Point 7: Figure captions are missing for all figures.
Response 7: We have revised in the manuscipt.
Reviewer 2 Report
In the manuscript titled “Analysis of influencing factors of apathy in patients with Parkinson's disease”, the authors have emphasized apathy, a non-motor deficit in Parkinson's disease (PD) patients which is a common neuropsychiatric symptom in patients with PD and adversely impacts the quality of life in PD patients. They evaluated the apathy score (AS) obtained from the patient groups enrolled based on the inclusion and exclusion criterion proposed by the Movement Disorder Society (MDS). Studies on 200 patients showed a significant (69%) incidence of apathetic cases and education and disease duration are influential factors of apathy. Further, the authors evaluated motor and non-motor symptoms between the two groups using various methods.
I appreciate the effort authors made in highlighting the neuropsychiatric aspects of PD which might be crucial to discovering ways to improve quality of life. Despite the commendable effort, this manuscript requires the following queries to be addressed to make this article appropriate for publication.
Major comments:
1. There is a very thin line between apathy and depression. Could you add some studies to show that the patients selected had no depression-like state?
2. Apathy is caused by several psycho‑behavioral factors even in non-PD persons. Age and gender-matched studies on healthy people could provide better insight into apathy in PD patients.
Minor comments:
1. Since there are many phycological factors that affect the state of mind in men and women differently, so separate representation of gender-based information in table 1 could be helpful in understanding gender-biased PD symptoms if any.
2. Graphical representation is eye-soothing and easy to understand and interpret information. Graphical representation for table 1 could be a better option for representation.
Author Response
Point1:There is a very thin line between apathy and depression. Could you add some studies to show that the patients selected had no depression-like state?
Response 1: Thanks for your reminding, we apply stepwise statistic method to remove the influence of depression such as other factors.
Point2 :Apathy is caused by several psycho‑behavioral factors even in non-PD persons. Age and gender-matched studies on healthy people could provide better insight into apathy in PD patients.
Response 2: Thanks for your reminding, Actually, there could be several other factors causes apathy in non-PD persons, so we chosen genser-matched PD-non-apathy and PD-apathy to achieve case-control study, and apply stepwise to remove confounding factors.
Round 2
Reviewer 2 Report
Thanks to the authors for revising the manuscript. I agree with the explanation authors have provided and significant changes make the manuscript in good shape.